# DiffProbe: Towards a Universal and Cross-Modality Adversarial Robustness Quantification Framework for Black-box Classifiers using Diffusion Models

## Abstract

Neural network classifiers have become popular, fundamentally transforming tasks across multiple data modalities such as image processing, natural language understanding, audio recognition, and more. Despite their widespread adoption, a critical challenge that persists is ensuring their robustness against adversarial attacks, which aim to deceive models through subtly modified inputs. This issue is particularly acute when considering interactions across different modalities, a facet that most current studies neglect. Addressing this gap, our paper introduces **DiffProbe**, the first unified black-box framework for adversarial robustness quantification using synthetically generated data from domain-specific diffusion models. **DiffProbe** stands out by seamlessly integrating off-the-shelf diffusion models to create a versatile and comprehensive framework tool suitable for a wide range of data types and adversarial scenarios. It is particularly designed for computational efficiency, making it ideal for evaluating black-box models and facilitating remote auditing with minimal requirement—only needing predictions from models on synthetic data. The robustness evaluation of **DiffProbe** is both theoretically sound and empirically robust, showing high consistency with real-world adversarial attack methods. We have extensively tested **DiffProbe** across various state-of-the-art classifiers and black-box APIs in domains including facial recognition, text, audio, video, and point cloud data. The results underscore its effectiveness in providing realistic and actionable insights into the adversarial robustness of systems, thus enhancing our understanding of adversarial vulnerabilities and aiding in the development of more secure AI systems across different modalities.

## 1 Introduction

Neural network classifiers are remarkably versatile, applying their capabilities across a wide array of data modalities and tasks including image recognition, natural language processing, audio analysis, and more. Despite their extensive applications, these models encounter significant risks from adversarial attacks which are designed to exploit model vulnerabilities and induce misclassification, known as adversarial robustness Chen & Liu (2023); Goodfellow et al. (2015). These risks underscore the need for robust methods to evaluate and enhance the security of neural network classifiers against such threats.

Adversarial robustness quantifies a model's ability to withstand malicious input perturbations aimed at causing misclassification Chen & Hsieh (2022). Evaluation methods for adversarial robustness are divided into two categories: *attack-dependent* and *attack-independent*. Attack-dependent methods challenge the model with the strongest attacks to gauge its performance under extreme conditions. Conversely, attack-independent methods offer a certified or estimated robustness score based on intrinsic properties of the model or data, without necessarily utilizing actual adversarial examples. Examples include neural network verification techniques W. & Zico (2018); Huan et al. (2018), randomized smoothing for certified defenses M. et al. (2019), and estimating local Lipschitz constants Tsui-Wei et al. (2018).

However, these methods encounter several issues. First, they often rely on aggregating local robustness scores from a finite set of samples, which can lead to biases and inefficiencies. Ideally, sampling from an infinitely

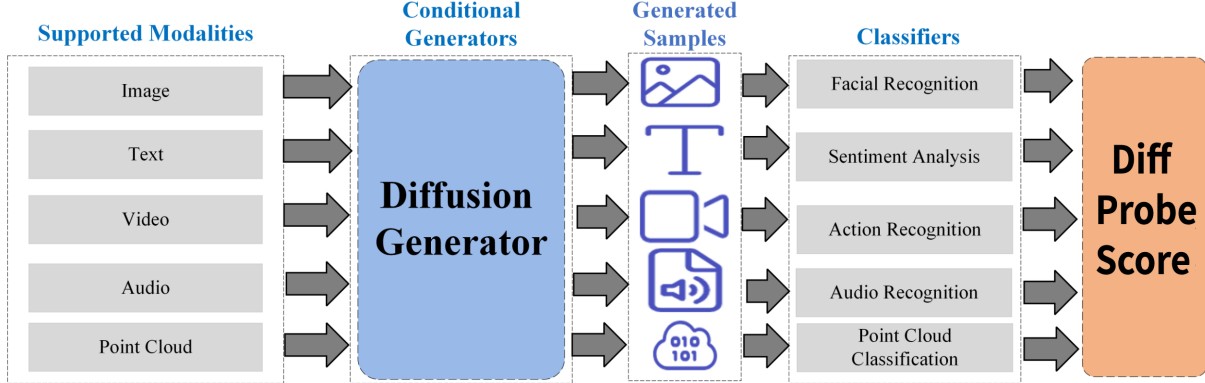

Figure 1: Overview of **DiffProbe**. It supports modalities (Image, Text, Video, Audio, Point Cloud), employs a Diffusion Generator to produce samples, which are then processed by classifiers to generate the DiffProbe Score use the prediction probability.

large dataset that represents the true data distribution would provide a comprehensive global robustness score. Second, many existing methods presuppose full knowledge of the model's architecture and parameters—a white-box setting—which is not feasible or realistic, especially in proprietary or sensitive applications. Third, much of the existing work is limited to image modalities, neglecting other critical data types such as audio, video, graphs, and point clouds. This narrow focus hinders the generalization and application of robustness techniques across diverse and increasingly important data modalities.

To address these challenges, we introduce **DiffProbe**, a novel framework that employs synthetic data generated from domain-specific diffusion models to quantify adversarial robustness in a black-box setting. DiffProbe is designed to evaluate global adversarial robustness across data modalities by using diffusion models as proxies for real data distributions, thereby overcoming the limitations associated with finite sample sizes. It operates efficiently in a black-box setting, requiring only access to the model's outputs, thereby preserving confidentiality of model internals and broadening its applicability. DiffProbe is computationally efficient, leveraging pre-trained generative models to avoid costs of training new models and necessitating only a single forward pass per sample, which enhances scalability and practicality for datasets.

Our main contributions are as follows:

- The development of DiffProbe, a unified black-box framework for adversarial robustness quantification using synthetic data from domain-specific diffusion models. DiffProbe's framework can be easily extended to new domains and diffusion models. Unlike prior work such as GREAT Score Li et al. (2023) (image-only, GAN-based) and DeepGRE Zhang et al. (2024) (image-only, white-box, GAN-based), DiffProbe operates in a purely black-box setting across five modalities using diffusion models with stronger theoretical convergence guarantees.

- The first comprehensive cross-modality adversarial robustness analysis covering domains such as facial recognition, text, audio, video, and point cloud data. These domains represent diverse data characteristics, including continuous and discrete data, structured and unstructured formats, and static, spatial-temporal, and temporal data. This selection also includes sequence and non-sequence data, offering a broad perspective on adversarial robustness.

- Validation of DiffProbe's effectiveness on state-of-art classifiers, demonstrating correlation with black-box attacks and consistent, reliable results at lower computational cost. We provide formal theoretical guarantees including score convergence bounds, distribution approximation bounds, and ranking preservation guarantees.

- Practical demonstrations of DiffProbe for the remote robustness evaluation of access-restricted systems, including audits of online black-box APIs for facial recognition and sentiment analysis. Using DiffProbe,

Table 1: Systematic comparison of adversarial robustness frameworks. Existing works are constrained by attack dependence, modality limitations, or black-box incompatibility. DiffProbe uniquely addresses all three challenges.

| Method | Attack-Dependent? | Modalities | Black-Box? |
|---|:---:|:---:|:---:|
| AutoAttack Croce & Hein (2020) | Yes (PGD) | Image | No |
| RobustBench Croce et al. (2021) | Yes (AutoAttack) | Image | No |
| GREAT Score Li et al. (2023) | No | Image | Yes |
| AdvGLUE Wang et al. (2021) | Yes (TextFooler) | Text | Partial |
| Imperio Qin et al. (2022) | Yes (Over-the-Air) | Audio | Yes |
| ShapeAdv Liu et al. (2022b) | Yes (Surface Perturb.) | Point Cloud | No |
| **DiffProbe (Ours)** | **No** | **Image, Text, Audio, Video, Point Cloud** | **Yes** |

we found, for example, that the Old group (generated facial images of elderly) is more robust than the Young group in the DEEPFACE API.

**Scope and Positioning.** DiffProbe is a *practical robustness quantification tool* that produces proxy robustness scores highly correlated with attack-based evaluations. It is designed as a fast, scalable first-pass assessment complementary to formal verification methods and attack-based evaluations, rather than a replacement for them. Its primary use case is remote auditing of black-box models and APIs where model internals are inaccessible.

## 2 Background and Related Works

### 2.1 Overview of Diffusion Models

Diffusion models generate synthetic data through iterative noise addition and denoising processes. Our framework leverages three key variants—**DDPMs** Ho et al. (2020), **DDIMs** Song et al. (2021a), and **Score SDEs** Song et al. (2021b) (see Appendix A.1 for detailed mechanisms).

**Core Mechanism:** In DDPMs, the forward process gradually perturbs data with Gaussian noise over $T$ steps:

$$q(\mathbf{x}_t|\mathbf{x}_{t-1}) = \mathcal{N}\left(\mathbf{x}_t; \sqrt{1-\beta_t}\mathbf{x}_{t-1}, \beta_t\mathbf{I}\right), \tag{1}$$

where $T$ denotes diffusion steps. The reverse process learns via a neural network $\mu_\theta$. DDIMs enhance sampling efficiency through non-Markovian dynamics, while Score SDEs generalize the process via stochastic differential equations.

### 2.2 Formal Local Robustness Guarantee and Estimation

Given a data sample $x$, a formal local robustness guarantee refers to a certified range of perturbations such that the top-1 class prediction of a model remains unchanged Hein & Andriushchenko (2017). For $\mathcal{L}_p$-norm ($p \geq 1$) bounded perturbations centered at $x$, this is called a certified radius $r$ such that any perturbation $\delta$ to $x$ within this radius (i.e., $\|\delta\|_p \leq r$) will have the same top-1 class prediction as $x$. The certification radius of $x$ is also a lower bound on the minimum perturbation necessary to flip the model's prediction.

### 2.3 Existing Adversarial Attack Benchmark

**Adversarial Robustness Evaluation.** Existing approaches to assess robustness primarily focus on optimizing adversarial attacks (e.g. TextFooler Jin et al. (2019)) or designing perturbation-resistant objectives. However, these efforts are fragmented across modalities and lack a unified evaluation paradigm. As summarized in Table 1, existing frameworks suffer from three critical shortcomings: **Attack Dependency**, where methods like AutoAttack Croce & Hein (2020) and AdvGLUE Wang et al. (2021) require crafting adversarial examples via optimization, making evaluations sensitive to attack hyperparameters. **Modality Fragmentation**

exists as benchmarks are siloed by data types—images (RobustBench Croce et al. (2021)), text (TextAttack Morris et al. (2020)), or audio (Imperio Qin et al. (2022))—with no cross-modal comparability. **Black-Box Incompatibility** arises as most methods (e.g., ShapeAdv Liu et al. (2022b)) demand white-box access, while real-world systems (e.g., DeepFace API) restrict model internals. While DiffProbe addresses these gaps through a fundamental shift: replacing attack-dependent evaluations with *generative synthetic testing*. Our key innovations include **Generative Robustness Scoring** that leverages diffusion models as synthetic data generators to approximate global robustness, eliminating attack optimization; **Modality-Agnostic Design** that enables unified evaluation across 5 modalities via domain-specific diffusion operators (Appendix A.1); and **Black-Box Auditing** that enables robustness assessment of restricted APIs (e.g., Amazon Sentiment Analysis) with only query access.

**Comparison with DeepGRE.** The most closely related work is DeepGRE Zhang et al. (2024), which also uses generative models for robustness evaluation. However, DiffProbe differs fundamentally in several aspects: (1) **Access model**: DeepGRE requires white-box access (gradients) for gradient-based robustness certificates, whereas DiffProbe operates in a purely black-box setting requiring only prediction outputs. (2) **Generator**: DeepGRE uses GAN-based generators, while DiffProbe leverages diffusion models with stronger theoretical convergence guarantees. (3) **Modalities**: DeepGRE is restricted to images, while DiffProbe supports five modalities (image, text, audio, video, point cloud). (4) **Attack dependency**: DeepGRE relies on gradient-based analysis, while DiffProbe is fully attack-free. (5) **API auditing**: DiffProbe uniquely supports remote auditing of black-box APIs, which DeepGRE cannot perform.

## 3 DiffProbe for Cross-modality Adversarial Robustness Analysis

**Terminology.** Throughout this paper, we use "cross-modality" to refer to the framework's ability to operate *across* multiple distinct data modalities (image, text, audio, video, and point cloud), rather than describing cross-modal retrieval or translation tasks (e.g., image retrieval from text queries). DiffProbe provides a unified scoring methodology that can be instantiated with domain-specific diffusion generators and classifiers for each modality.

**Robust Accuracy.** We define Robust Accuracy (RA) as the classification accuracy on adversarially perturbed samples: $\text{RA} = \frac{1}{N} \sum_{i=1}^{N} \mathbb{1}[\phi(x_i + \delta_i^*) = y_i]$, where $\delta_i^*$ is the adversarial perturbation found by the attack method, and $\phi$ is the classifier. RA serves as our ground-truth metric for validating DiffProbe scores.

In this section, we begin by introducing the adversarial robustness qualification metric used in this framework in Section 3.1. Then, we describe the algorithm for DiffProbe in Section 3.2. In Section 3.2, we explain the computational complexities. In Section 3.3, we explain the motivations on using diffusion models for robustness evaluation. Finally, we present the various domains, classifiers, and other components involved in DiffProbe in Section 3.5. Table 2 summarizes all the models, classifiers, and attacks employed in this framework.

Table 2: A Summary of All the Models and Datasets Used in DiffProbe.

| Modality | Diffusion Generator | Task | Classifiers/APIs | Adversarial Attack | Dataset |
|---|---|---|---|---|---|
| Image | Stable Diffusion | Facial Recognition | DeepFace | Square Attack | LAION-5B |
| Video | VideoCrafer | Action Recognition | Slowfast | Geo-TRAP Attack | UCF101 |
| | | | I3D | | |
| | | | C3D | | |
| Text | LatentOps | Sentiment Analysis | Xlnet | Textbugger Attack | Yelps Review |
| | | | Bert | | |
| | | | Microsoft | | |
| | | | Amazon | | |
| Audio | Diffwave | Audio Recognition | Liquid-S4 | Kenansville Attack | Speech Commands |
| | | | S4 | | |
| Point Cloud | PVD | Shape Classification | HyCoRe | Shape-Invariant Attack | ShapeNetCore / ModelNet40 |
| | | | PointNet | | |

### 3.1 Cross-Modal Generalization for Robustness Quantification

The methodological framework draws inspiration from the generality of Transformer architectures to process heterogeneous data through unified computation graphs. Building upon GREAT Score's theoretical foundation Li et al. (2023), we achieve cross-modal generalization via three synergistic components, while strictly utilizing pre-trained diffusion models without architectural modifications.

First, inspired by how Transformers use positional encodings for modality-specific processing, we adapt diffusion operators to extend GREAT Score, originally designed for images, to other modalities. For text, masked language modeling ensures semantic coherence during perturbation generation; for point clouds, graph diffusion with geometric constraints preserves topology. Audio processing utilizes spectral-domain diffusion to respect acoustic properties. These adaptations seamlessly integrate with existing diffusion generators, eliminating the need for model retraining.

Second, mirroring the cross-modal embedding projection in multimodal Transformers, we utilize a conditioning mechanism that maps diverse prompts to the latent space of pre-trained generators. The text-to-image pathway employs CLIP's joint embedding space to align linguistic descriptions with visual concepts, whereas shape-to-point-cloud conversion utilizes PCA-based geometric encoding to preserve structural integrity. This approach enables unified conditional generation across modalities through frozen diffusion decoders $D_m$, with our innovation residing in the input conditioning paradigm rather than the generative models themselves.

Third, we introduce a normalized robustness metric that parallels the layer normalization mechanism in Transformers, while adapting to any modality diffusion models. The core scoring function is formally defined as:

$$
\begin{aligned}
\mathcal{S}_m &= \sqrt{\frac{\pi}{2}} \cdot \mathbb{E}_{y \sim U(\{1,2,\ldots,K\});\ z \sim \mathcal{N}(0,I)}\ S(y,z|G,\phi) \\
&\text{and } S := \max\left(\phi_y(G(z|y)) - \max_{k \neq y} \phi_k(G(z|y)), 0\right)
\end{aligned}
\tag{2}
$$

where $n$ is the number of generated samples, $\phi : [0,1]^d \mapsto [0,1]^K$ is a $K$-way classifier and $\phi_k(\cdot)$ is the predicted likelihood of class $k$, $\{z_1,\ldots,z_n\} \sim \mathcal{N}(0,I)$, $y_i \in \{1,2,\ldots,K\}$ is the randomly chosen class label for conditional generation ($y \sim U(\{1,2,\ldots,K\})$ denotes uniform sampling from $\{1,2,\ldots,K\}$), and $G(z_i|y_i)$ is the generated sample.

**Intuitive Interpretation.** The local score $S(y,z|G,\phi) = \max(\phi_y(G(z|y)) - \max_{k \neq y} \phi_k(G(z|y)), 0)$ measures the *confidence margin* of the classifier on a generated sample: the gap between the probability assigned to the correct class and the highest probability assigned to any incorrect class. A larger margin indicates that the classifier is more confident and thus more likely to withstand adversarial perturbations. The $\max(\cdot, 0)$ truncation ensures that misclassified samples contribute zero to the score. The $\sqrt{\pi/2}$ scaling factor is a bias correction arising from the half-normal distribution, which ensures that the sample mean provides an unbiased estimate of the expected confidence margin. The DiffProbe Score $\mathcal{S}_m$ thus quantifies the *global adversarial robustness* of a classifier by averaging this margin over synthetically generated data. This formulation builds upon the theoretical foundation of GREAT Score Li et al. (2023), which proves that this scoring function provides a lower bound on true global robustness. DiffProbe extends GREAT Score by: (a) replacing GAN generators with diffusion models that have stronger theoretical convergence guarantees Chen et al. (2024; 2022); (b) introducing cross-modal conditioning mechanisms for five modalities; and (c) validating the framework across diverse data types beyond images.

### 3.2 Algorithm for DiffProbe

Algorithm 1 details the computation of the DiffProbe score using a sample mean estimator approach. The DiffProbe framework is designed for scalability and can be easily extended to new domains, diffusion models, and classifiers.

We analyze the computation complexity of DiffProbe as follows. Consider $N_S$ samples. Each sample is generated by the diffusion generator $G(\cdot)$ and evaluated by the classifier $\phi(\cdot)$ to compute local scores $s(\cdot)$.

---

**Algorithm 1:** DiffProbe Score Calculation

---

**Input:** Classifier $\phi(\cdot)$ with $K$ classes,
  Conditional diffusion generator $G(\cdot)$,
  Number of samples $N_S$.
**Output:** Computed DiffProbe Score $\widehat{\mathcal{S}}_m$
**for** $i \leftarrow 1$ **to** $N_S$ **do**
  Sample a random class label $y$ uniformly from $\{1, 2, \ldots, K\}$
  Sample a latent vector $z$ from a standard normal distribution $z \sim \mathcal{N}(0, I)$
  Generate a sample $x = G(z|y)$ conditioned on class $y$
  Use the classifier $\phi$ to predict the class probabilities $\{\phi_k(x)\}_{k=1}^K$
  Calculate the local score $s^{(i)} \leftarrow \max\left(\phi_y(x) - \max_{k \neq y} \phi_k(x), 0\right)$
**end**
Compute the final score $\widehat{\mathcal{S}}_m \leftarrow \sqrt{\frac{\pi}{2}} \cdot \left(\frac{1}{N_S} \sum_{i=1}^{N_S} s^{(i)}\right)$

---

The total computational cost depends on the number of samples $N_S$, the classifier $\phi(\cdot)$, and the generator $G(\cdot)$. The time complexity $T(\Omega)$ for the DiffProbe score is:

$$T(\Omega) = O\left(N_S \times (T(\phi(\cdot)) + T(G(\cdot)))\right) \tag{3}$$

where $T(\phi(\cdot))$ is the classifier's inference time and $T(G(\cdot))$ is the sample generation time.

In essence, the computational cost scales with the number of samples $N_S$, combining the inference complexities of both the classifier and the generator.

### 3.3 Motivations for Using Generative Diffusion Models in Robustness Evaluation

Conventional robustness evaluation methods often rely on limited test sets, which may not fully capture the true data distribution of the evaluated datasets. This limitation can lead to incomplete or potentially biased assessments of model robustness. To address this critical issue, we propose leveraging generative diffusion models as a more comprehensive proxy for the underlying true data distribution.

Our choice of diffusion models is motivated by their ability to generate samples, exploring regions of the data space that are underrepresented in finite test sets. This expanded coverage allows for a more thorough evaluation of model robustness across a broader spectrum of potential inputs.

The use of diffusion models is also supported by recent theoretical advancements in the field of generative modeling. Notably, studies by Chen et al. (2024) and Chen et al. (2022) have established rigorous theoretical foundations for diffusion models. These works demonstrate that, under certain conditions, diffusion generators can asymptotically approximate the real data distribution with high fidelity, offering convergence guarantees in learning true data distributions.

### 3.4 Theoretical Guarantees

We provide formal guarantees for the DiffProbe score estimator.

**Theorem 1** (Score Convergence). *Let $s^{(i)}$ be the local score for sample $i$, bounded in $[0, C]$ where $C \leq 1$. The sample mean estimator $\widehat{\mathcal{S}}_m = \sqrt{\pi/2} \cdot \frac{1}{N_S} \sum_{i=1}^{N_S} s^{(i)}$ satisfies:*

$$P\left(|\widehat{\mathcal{S}}_m - \mathcal{S}_m| \geq \epsilon\right) \leq 2\exp\left(-\frac{2N_S \epsilon^2}{\pi C^2/2}\right) \tag{4}$$

*where $\mathcal{S}_m = \sqrt{\pi/2} \cdot \mathbb{E}[s^{(i)}]$ is the true DiffProbe Score.*

*Proof.* Since each $s^{(i)} \in [0, C]$ and the samples are i.i.d., the result follows directly from Hoeffding's inequality applied to the scaled sum $\sqrt{\pi/2} \cdot \frac{1}{N_S} \sum_i s^{(i)}$. $\square$

**Theorem 2** (Distribution Approximation Bound). *Let $p_G$ denote the distribution of the diffusion generator and $p_{data}$ the true data distribution. If the scoring function $S(y, z|G, \phi)$ is L-Lipschitz with respect to the total variation (TV) distance, then:*

$$|\mathcal{S}_m(p_G) - \mathcal{S}_m(p_{data})| \leq L \cdot TV(p_G, p_{data}) \tag{5}$$

*This bound ensures that the score error diminishes as the generator quality improves, which is guaranteed by the convergence properties of diffusion models Chen et al. (2024; 2022).*

**Proposition 3** (Ranking Preservation). *Let $\phi_A$ and $\phi_B$ be two classifiers with true DiffProbe Scores $\mathcal{S}_m^A > \mathcal{S}_m^B$. If the score gap $\Delta = \mathcal{S}_m^A - \mathcal{S}_m^B > 0$ satisfies $\Delta > 2L \cdot TV(p_G, p_{data}) + 2\epsilon$, then with probability at least $1 - 4\exp(-2N_S\epsilon^2/(\pi C^2/2))$, the estimated scores preserve the ranking: $\widehat{\mathcal{S}}_m^A > \widehat{\mathcal{S}}_m^B$.*

This result formalizes the empirical observation that DiffProbe rankings remain consistent even with approximate generators, provided the score gap between classifiers is sufficiently large relative to the generator approximation error.

### 3.5 Framework Composition

We propose a comprehensive framework to evaluate adversarial robustness that includes **5** distinct domains, **12** classifiers, **5** diffusion models (DMs), and **5** attack methods. The domains encompass a variety of applications such as face recognition and sentiment analysis. This setup provides assessment of model performance and robustness in real-world scenarios.

## 4 Experimental Results

### 4.1 Experimental Setup

**Datasets.** We evaluate our approach on five different domains: facial recognition, text, audio, graph, video, and point cloud. For each domain, we use a publicly available dataset that is widely used in the literature. Specifically, we use UCF101 Soomro et al. (2012) for video, Yelp Review Shen et al. (2017) for text, ShapeNetCore Chang et al. (2015) and ModelNet40 Wu et al. (2015) for point cloud, Speech Commands Dataset Warden (2018) for audio.

**Generative Models and Classifiers.** For each domain, we select a pretrained diffusion model to generate data samples from the corresponding dataset. To benchmark the quality of the generated samples, we select at least two classifiers for each domain.

**Framework Implementation.** We implement the framework as described in Section 3.2, where we apply a sigmoid to the logits of each classifier under evaluation to ensure that the model output of each dimension is within [0,1], as suggested by Li et al. (2023). We use 500 samples per class from the diffusion to compute the framework score. For conditional generation, we use 500 samples per class from the diffusion model to compute the framework score. For audio generation, since it is unconditional, we sample 5000 samples together for a total of 10 classes and select 300 samples per class with the highest quality.

**Comparative methods.** We compare effectiveness of our framework score in two aspects: time-efficiency and high correlation with robust accuracy under attack. For each classifier, we run a black-box adversarial attack and report the accuracy and running time. Our objective is to show that classifiers with higher framework scores are more robust to adversarial attacks. The details of the parameter settings for the classifiers and the adversarial attack are listed in Appendix A.3.

**Compute Resources.** All our experiments were run on 2 GTX 3090 GPUs with 24GB RAM.

The following sections break down the implementation and results of DiffProbe by domain.

### 4.2 Image

**Text-to-Image Generative Diffusion Model.** For the image generation task, we utilize the Stable Diffusion model Rombach et al. (2021). This model synthesizes realistic and diverse images from input text.

Stable Diffusion employs the DDIM mechanism in latent space, leveraging powerful pre-trained denoising autoencoders. This allows the model to transfer input data samples into latent space and perform the diffusion process efficiently, making it feasible to train on limited computational resources.

**Dataset.** Stable Diffusion was trained on image-text pairs from the LAION-5B dataset Schuhmann et al. (2022). This dataset includes 5 billion image-text pairs, classified based on language and filtered into separate datasets by resolution, predicted probability, and predicted "aesthetic" score.

**Text prompts and conditions for the generator.** For image generators, we specify the input text prompts with given attributes and gender, e.g., old man, young woman, with eyeglasses, and without eyeglasses. For example, an input prompt might be "a natural and professional photograph of a young man's detailed face." For the female generation, we change "man" to "woman." Further details and examples are provided in the Appendix A.2.

**Facial Recognition and Classifiers.** Facial recognition involves identifying or verifying a person's identity using their facial features. For our experiments, we use an online facial recognition API for gender classification, capable of identifying gender from an image. Specifically, we evaluate the DeepFace API Serengil & Ozpinar (2020), a lightweight face recognition and attribute tool.

**Adversarial Attack for Facial Recognition.** We use the Square Attack to evaluate the robustness of the face recognition API. The Square Attack Andriushchenko et al. (2020) is a score-based black-box adversarial attack that does not rely on local gradient information, thereby bypassing the gradient requirement. It employs a randomized search scheme that selects localized square updates at random positions, ensuring that at each iteration, the perturbation is approximately at the boundary of the feasible set.

**DiffProbe Score v.s. Robust Accuracy.** Using DiffProbe, we assessed group robustness of the DEEPFACE gender classification API. Table 4 summarizes our findings, revealing a significant robustness disparity between the Old and Young groups for the DEEPFACE API, whereas other APIs exhibit more consistent scores. Moreover, the With Eyeglasses group achieved a higher robustness score than the Without Eyeglasses group, suggesting that eyeglasses might serve as a spurious feature affecting robustness. To validate our evaluation, we compared the DiffProbe Score with the results from the black-box square attack Andriushchenko et al. (2020), using $\epsilon = 2$ and 100 queries on the DEEPFACE API. For both Age and Eyeglasses groups (Old vs. Young and With vs. Without Eyeglasses), we observed a consistent correlation between a higher DiffProbe Score and improved robust accuracy.

## 4.3 Video

**Text-to-Video Generative Diffusion Model.** Generating high-quality videos is challenging due to spatio-temporal continuity of frames. To conditionally generate videos with given labels, we integrate the toolkit VideoCrafter [1], a text-to-video generative diffusion model derived from Stable Diffusion.

**Dataset.** We choose UCF101 Soomro et al. (2012) as the dataset because it has the largest variety of actions with 13,320 videos from 101 action categories.

**Action Recognition and Classifiers.** Action recognition aims to identify human actions and interactions in video. We choose three classifiers: SlowFast Feichtenhofer et al. (2019), I3D Carreira & Zisserman (2017), and C3D Tran et al. (2015). SlowFast operates at two different frame rates to capture both slow and fast motion. I3D inflates 2D convolutions to 3D, enabling spatiotemporal feature learning. C3D uses 3D convolutional networks to model temporal dynamics in video data.

**Text prompts for generator.** For video generator, we specify the text prompts with 5 labels: "play basketball", "haircut", "play violin", "pull up" and "blow dry hair". To improve the quality of the generated examples, we tune the prompts to one sentence. The detailed prompts we used for the generator and the generated examples are shown in Appendix A.2.

**Adversarial Attack for Action Recognition on Videos.** We use Geo-Trap Li et al. (2021), a black-box adversarial attack to evaluate the classifiers. Query-efficient black-box attacks rely on estimated gradients in

---

[1] https://github.com/VideoCrafter/VideoCrafter

the searching space. These gradients are estimated by searching for 'directions' that maximize the probability of the victim model misclassifying the crafted inputs. Geo-Trap employs geometric transformation operations to reduce the search space for effective gradients.

**DiffProbe Score v.s. Robust Accuracy.** For video data, the Table 3 shows the model ranking: Slowfast > i3d > c3d. Slowfast achieves the highest robust accuracy (10.8%) and the highest DiffProbe score (0.0802). i3d follows with a robust accuracy of 8.5% and a DiffProbe score of 0.0503. c3d has the lowest robust accuracy (3.2%) and score (0.0001). This ranking shows a strong positive correlation between DiffProbe Scores and robust accuracy in video models, highlighting the effectiveness of DiffProbe as a robustness metric.

## 4.4 Text

**Generative Diffusion Model for Sentences.** To generate sentences with specified sentiment attributes , we use the LatentOps framework Liu et al. (2022a). LatentOps allows for control operations within the continuous latent space of text using Score SDEs. First, it uses pre-trained language models (e.g., GPT-2 Radford et al. (2019)) to convert text into a continuous representation. Then, sentiment attributes are applied to these latent vectors. LatentOps samples the desired latent vectors $z_0$ by solving ODEs that reverse the diffusion process. Finally, the sampled $z_0$ is input to a GPT-2 decoder to generate the target text sequence.

| Modality | Model 1 / | Model 2 / | Model 3 | Robust Accuracy (%) | DiffProbe Score |
|---|---|---|---|---|---|
| Video | Slowfast | i3d | c3d | 10.8 / 6.16 / 3.2 | 0.0802 / 0.0483 / 0.0001 |
| Text | BERT | Xlnet | - | 37.7 / 46.8 | 0.0273 / 0.0428 |
| Point Cloud | HyCoRe | Pointnet | - | 61.8 / 90.1 | 0.197 / 0.560 |
| Audio | Liquid-S4 | S4 | - | 45.6 / 31.5 | 0.095 / 0.057 |
| Sentiment Analysis API | Amazon | Microsoft | - | - | 0.993 / 0.969 |

Table 3: DiffProbe vs. Robust Accuracy under various attacks across different data modalities

**Sentiment Analysis and Classifiers.** Sentiment analysis identifies sentiments in text, such as positive or negative opinions. We evaluate two models: XlNet Yang et al. (2019) and BERT Devlin et al. (2018). XlNet uses a generalized autoregressive pre-training method for bidirectional context learning by maximizing expected likelihood over all input sequence permutations. BERT pretrains deep bidirectional representations by conditioning on both left and right context across all layers.

**Dataset.** We evaluate our methods on Yelp Review dataset Shen et al. (2017), which provides information on customer opinions and preferences, as well as business attributes.

**Text Prompts for Generator.** We specify sentiment attributes as either negative or positive .

**Adversarial Attacks on Sentiment Analysis Models.** Adversarial attacks in text aim to manipulate words or characters to create adversarial examples that mislead sentiment analysis models. We assess robustness using TextAttack Morris et al. (2020), a Python framework for adversarial attacks on NLP models. We use a black-box attack method called TextBugger Li et al. (2018), which identifies important words using a scoring function based on the classification result and then alters them using a selection algorithm.

**DiffProbe Score v.s. Robust Accuracy.** For text data, Table 3 shows the results for the BERT and XlNet models. XlNet outperformed BERT with a robust accuracy of 46.8% and a DiffProbe Score of 0.0428, compared to BERT's 37.7% robust accuracy and 0.0273 score. This supports the correlation between DiffProbe Scores and robust accuracy in text models.

**Online Sentiment Analysis APIs.** We utilize two online sentiment analysis APIs from Amazon[2] and Microsoft[3]. These APIs accept text input and return sentiment labels (positive, negative, neutral) along with corresponding sentiment scores. Our results for text, as shown in Table 3, indicate that the Amazon sentiment analysis API is more robust compared to the Microsoft counterpart.

---

[2]https://aws.amazon.com/cn/comprehend/
[3]https://azure.microsoft.com/en-us/products/cognitive-services

### 4.5 Point Cloud

**Generative Diffusion Model for Point Cloud.** A point cloud is a collection of discrete points in 3D space, each defined by its coordinates. Point cloud generation aims to create sets of points forming shapes. Luo & Hu (2021) present a probabilistic model for point clouds, inspired by non-equilibrium thermodynamics. They conceptualize points as particles in a thermodynamic system interacting with a heat bath, diffusing from an initial to a noise distribution. The reverse diffusion process, modeled as a Markov chain conditioned on a shape latent, transforms the noise into the desired shape.

**Dataset.** This study examines two datasets: ShapeNetCore Chang et al. (2015) and ModelNet40 Wu et al. (2015), utilized for generation and classification tasks respectively. ShapeNetCore, features 3D models with verified annotations. ModelNet40, contains synthetic object point clouds and serves as a benchmark due to its diverse categories and shapes. It comprises 12,311 CAD-generated meshes across 40 categories.

**Conditions for Generator.** For point cloud generators, we specify the shape as chairs or airplanes.

**Classifiers for Point Cloud Classification.** To assess the robustness of our approach, we employ Hycore Montanaro et al. (2022) and PointNet Qi et al. (2017). Hycore reconceptualizes the compositionality of point clouds by embedding the features of a point cloud classifier in hyperbolic space, a non-Euclidean space that effectively represents the hierarchical tree-like structure of point clouds. PointNet addresses the rotation problem in point clouds using a spatial transformation network (STN). Given that the classifiers were originally trained on 40 labels, while our generator focuses solely on generating chairs and airplanes, we retain only the predicted logits for these two labels.

**Adversarial Attack.** Shape-invariant attacks Huang et al. (2022) are a novel type of adversarial perturbation that maintain the shape and visual appearance of the original point clouds while deceiving target models. These attacks are generated with an explicit constraint that limits perturbations along the normal vectors of the point cloud surface, ensuring that the perturbations align with the shape surface and do not introduce noticeable outliers or deformations.

**DiffProbe Score v.s. RA.** Table 3 provides an insightful comparison of point cloud models under attack. The PointNet model stands out with an impressive ra of 90.1% and a corresponding DiffProbe Score of 0.560, highlighting its robustness in handling adversarial conditions. On the other hand, the HyCoRe model, with a robust accuracy of 61.8% and a DiffProbe Score of 0.197, shows a significant gap in performance.

### 4.6 Audio

**Dataset.** The Speech Commands Dataset Warden (2018) contains one-second audio clips with a spoken English word or background noise. These words come from a limited set of commands and are spoken by diverse speakers, making it suitable for speech recognition and audio synthesis.

**Generative Diffusion Model.** DiffWave Kong et al. (2020) is probabilistic diffusion model for raw audio synthesis based on principles of DDPM. Unlike autoregressive models, DiffWave transforms white noise into structured waveforms through a Markov chain with fixed number of synthesis steps. We use a classifier trained on SC09 Warden (2018) dataset provided by the authors of DiffWave to assign labels to generated samples, and select 500 audio samples for each label.

**Conditions for generator.** For audio generators, we specify the generated digits to be 0-9.

**Classifiers.** For audio classification, we utilize Liquid-S4 Hasani et al. (2022) and S4 Gu et al. (2022) models, both structural state-space models. State-space models are neural networks that learn latent representations of sequential data by applying linear transformations and nonlinear activations to hidden states. Liquid-S4 is an advanced variant that incorporates a liquid time-constant (LTC) state-space model as its core. LTC neural networks are causal continuous-time networks featuring an input-dependent state transition module, allowing them to adapt dynamically to inputs during inference.

**Adversarial Attack on Audio Classifiers.** We employ the Kenansville attack Abdullah et al. (2021), an adversarial attack targeting speech recognition systems. This black-box attack modifies only a few frames of audio to induce misclassification. The Kenansville attack operates by removing the lowest-power spectral

Table 4: DiffBench v.s. Accuracy under square attack Andriushchenko et al. (2020)

| DEEPFACE | Old | Young | With Eyeglasses | Without Eyeglasses |
|---|---|---|---|---|
| Square Attack | 40.20% | 39.20% | 43.20% | 42.80% |
| Score | 0.917 | 0.883 | 0.909 | 0.8143 |

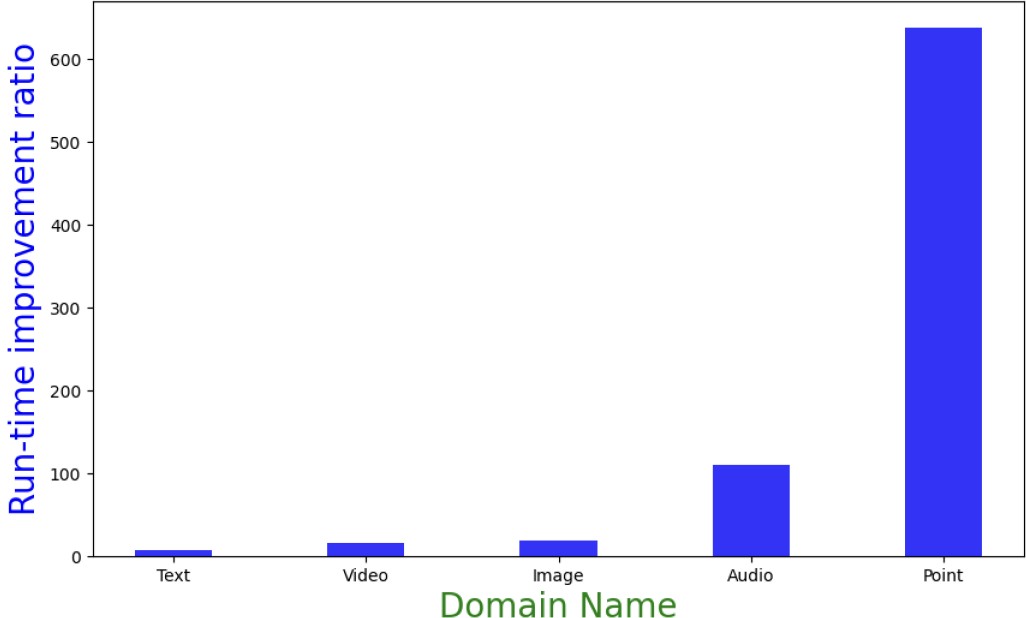

Figure 2: Run-time improvement over Adversarial Attack on generated samples for each domain.

density components of the input audio, effectively eliminating parts of the signal that are less perceptible to the human ear but crucial for speech recognition systems to correctly identify words.

**DiffProbe Score vs. Robust Accuracy.** For audio data, Table 3 reveals significant differences between the Liquid-S4 and S4 models. The advanced Liquid-S4 model achieves a robust accuracy of 45.6% and a correspondingly high DiffProbe Score of 0.095, whereas the standard S4 model records a considerably lower robust accuracy of 31.5% and a DiffProbe Score of 0.057. This substantial disparity clearly highlights the effectiveness of the advanced module in Liquid-S4, which incorporates a sophisticated liquid time-constant (LTC) state-space model at its core, demonstrating its superior performance in challenging audio processing tasks.

### 4.7 Adversarial Robustness Analysis Across Domains

In this section, we examine the adversarial robustness of models across various domains, focusing separately on continuous and discrete data types. Continuous domains include video and audio, while discrete domains encompass text and point cloud. Our analysis indicates that robustness scores generally align with robust accuracy within each data type. For instance, within continuous domains, audio classifiers tend to achieve higher robustness scores and robust accuracy compared to video classifiers. Similarly, within discrete domains, point cloud classifiers exhibit high robustness and robust accuracy, demonstrating strong resistance to adversarial attacks, comparing to text classifiers.

## 4.8 Runtime Analysis

Figure 2 illustrates the runtime efficiency of DiffProbe compared to a black-box adversarial attack method applied to the same set of generated data samples. We present the ratio of their average runtimes per sample, with exact times detailed in Appendix A.4. The results show a significant speedup, ranging from 8 to 600 times faster, demonstrating the substantial computational efficiency of our framework. This improvement highlights DiffProbe's value for large-scale testing scenarios.

## 5 Conclusions

In this paper, we present DiffProbe, a versatile and computationally efficient black-box adversarial robustness quantification framework. DiffProbe uses off-the-shelf generative diffusion models to facilitate cross-modality robustness analysis. Our framework covers state-of-the-art classifiers and black-box APIs from diverse domains, including image, text, audio, video, and point cloud, with the potential to extend to other emerging modalities and future AI applications. The computation of DiffProbe is lightweight and scalable as it only requires access to the model predictions on the generated data samples. In comparison to black-box adversarial attacks and real-world APIs, DiffProbe shows high consistency, demonstrating the efficiency in remote model auditing. Besides, DiffProbe only requires little time comparing to black-box adversarial attack. Our study suggests a new paradigm toward a unified cross-modality robustness quantification framework.

**Limitations and Societal Impacts.** One potential limitation of our study is that we did not conduct experiments on multi-modal models. Instead, we focused on evaluating each domain separately. In future work, we plan to extend our approach to multi-modal scenarios to comprehensively assess the adversarial robustness of models that process and integrate data from multiple modalities simultaneously. We do not see any ethical or negative impacts in our work.

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

## A  Appendix

### A.1  Mathematical Foundations of Diffusion Models

This section details the mathematical frameworks of three core diffusion paradigms used in DiffProbe: Denoising Diffusion Probabilistic Models (DDPMs), Denoising Diffusion Implicit Models (DDIMs), and Score-Based Generative Modeling via Stochastic Differential Equations (Score SDEs).

**Denoising Diffusion Probabilistic Models (DDPMs)**

The forward process progressively adds Gaussian noise to data $x_0$ over $T$ steps:

$$q\left(\boldsymbol{x}_t \mid \boldsymbol{x}_0\right) = \mathcal{N}\left(\boldsymbol{x}_t; \sqrt{\bar{\alpha}_t}\boldsymbol{x}_0, (1 - \bar{\alpha}_t)\,\boldsymbol{I}\right), \tag{6}$$

where $\alpha_t := 1 - \beta_t$, $\bar{\alpha}_t := \prod_{s=1}^{t} \alpha_s$, and $\beta_t$ is the noise schedule. A closed-form sampling rule enables direct generation of $x_t$ from $x_0$:

$$\mathbf{x}_t = \sqrt{\bar{\alpha}_t}\mathbf{x}_0 + \sqrt{1 - \bar{\alpha}_t}\boldsymbol{\epsilon}, \quad \boldsymbol{\epsilon} \sim \mathcal{N}(0, \boldsymbol{I}). \tag{7}$$

The reverse process learns to denoise via a neural network $\epsilon_\theta$:

$$p_\theta(\mathbf{x}_{t-1} \mid \mathbf{x}_t) = \mathcal{N}\left(\mathbf{x}_{t-1}; \mu_\theta(\mathbf{x}_t, t), \Sigma_\theta(\mathbf{x}_t, t)\right). \tag{8}$$

**Denoising Diffusion Implicit Models (DDIMs)**

DDIMs generalize DDPMs through non-Markovian dynamics. The forward process is redefined as:

$$q_\sigma\left(\mathbf{x}_{t-1} \mid \mathbf{x}_t, \mathbf{x}_0\right) = \mathcal{N}\left(\mathbf{x}_{t-1}; \sqrt{\bar{\alpha}_{t-1}}\mathbf{x}_0\right.$$
$$\left. + \sqrt{1 - \bar{\alpha}_{t-1} - \sigma_t^2}\frac{\mathbf{x}_t - \sqrt{\bar{\alpha}_t}\mathbf{x}_0}{\sqrt{1 - \bar{\alpha}_t}}, \sigma_t^2\boldsymbol{I}\right), \tag{9}$$

leading to an accelerated sampling rule:

$$\boldsymbol{x}_{t-1} = \sqrt{\alpha_{t-1}} \underbrace{\left(\frac{\boldsymbol{x}_t - \sqrt{1 - \alpha_t}\epsilon_\theta^{(t)}\left(\boldsymbol{x}_t\right)}{\sqrt{\alpha_t}}\right)}_{\text{Predicted } \boldsymbol{x}_0}$$
$$+ \underbrace{\sqrt{1 - \alpha_{t-1} - \sigma_t^2} \cdot \epsilon_\theta^{(t)}\left(\boldsymbol{x}_t\right)}_{\text{Direction to } \boldsymbol{x}_t} + \underbrace{\sigma_t \epsilon_t}_{\text{Random noise}},$$
$$\epsilon_t \sim \mathcal{N}(0, \boldsymbol{I}). \tag{10}$$

**Score-Based Generative Modeling (Score SDEs)**

Score SDEs unify diffusion via continuous-time stochastic differential equations. The forward process perturbs data with:

$$\mathrm{d}\mathbf{x} = \mathbf{f}(\mathbf{x}, t)\mathrm{d}t + g(t)\mathrm{d}\mathbf{w}, \tag{11}$$

where $\mathbf{w}$ is a Wiener process. Sampling reverses this SDE:

$$\mathrm{d}\mathbf{x} = \left[\mathbf{f}(\mathbf{x}, t) - g(t)^2 \nabla_\mathbf{x} \log p_t(\mathbf{x})\right] \mathrm{d}t + g(t)\mathrm{d}\bar{\mathbf{w}}, \tag{12}$$

where $\nabla_\mathbf{x} \log p_t(\mathbf{x})$ is the score function, and $\bar{\mathbf{w}}$ denotes reverse-time Wiener dynamics.

Table 5: Detailed Clock Time for each evaluation and adversarial attacks.

| Model Name | Domain | Score Evaluation | Adversarial Attack |
|---|---|---|---|
| Slowfast | Video | 142s | 3852s |
| C3d | | 113s | 600s |
| I3d | | 122s | 960s |
| Bert | Text | 57s | 356s |
| Xlnet | | 113s | 960s |
| Liquid-s4 | Audio | 206s | 23678s |
| S4 | | 234s | 23794s |
| Hycore | Point Cloud | 23s | 16776s |
| PointNet | | 8s | 6598.8s |
| DeepFace | Image | 240s | 4664s |

### A.2 Detailed text prompts and Conditions

Since simply entering the label information as text prompts for stable-diffusion and videocrafter will not produce a good result. We fine-tune the labels as follows: A woman is blow-drying her hair, A man is doing a chin-up, A man is getting a haircut, A man is playing the violin, A man is playing basketball.

For videos: "A Woman is Blowing Dry Hair", "A Human is Doing a Pull Up", "A Human is Getting a Haircut", "A Human is Playing the Violin" and "A Human is Playing Basketball".

For images: "a natural and professional photograph of a young man's detailed face", "a natural and professional photograph of an old man's detailed face", "a natural and professional photograph of a man's detailed face without glasses", "a natural and professional photograph of a man's detailed face with glasses". For the female generation, we simply change the man to a woman. Note that for the old woman generation, we change "old woman" to "60 years old" due to the low accuracy of the classification.

### A.3 Parameter Settings

For classifiers of audio, point cloud, we use the training script provided by the original authors to get the checkpoint. For text classifiers, we load the xlnet_large_cased and bert_base_uncased. For video classifiers, we load the checkpoints provided by Geo-Trap, the attack repository. For text and image APIs, we refer to deepface and eden ai to load them.

For adversarial attack, for video, we use the other parameters provided in the repository, changing the number of evaluation videos to 2500 and the maximum number of queries from 60000 to 600 to confirm that For text adversarial attack, we use the default settings used in the original Textattack. For point cloud adversarial attack, we use the same attack parameters. For audio attack, we use rid width as 8 to attack the audios.

### A.4 True running time

Table 5 shows the detailed clock time for each evaluation and adversarial attack.

## B Experiment Code

We provided the code repository in attached code with `https://anonymous.4open.science/r/DiffProbe_code-D35D/`

### B.1 Proofment for GREAT Score.

Please refer to GREAT Score Li et al. (2023) for the complete proof.

