# OpenReview forum: "DiffProbe: Towards a Universal and Cross-Modality Adversarial Robustness Quantification Framework for Black-box Classifiers using Diffusion Models"
_TMLR — Rejected by TMLR_

### Review · Reviewer_ai8U · 2026-01-01

**Summary Of Contributions:**

The paper targets a unified cross-modality robustness quantification setting under black-box access. The framework requires only model outputs on generated samples (i.e., no gradients/parameters), aligning well with remote auditing/API evaluation use cases. The approach leverages pre-trained diffusion generators instead of training new generative models, which improves accessibility and lowers deployment overhead. The score computation is straightforward, and the paper provides an explicit complexity discussion.

**Audience:**

Yes

**Audience Explanation:**

TMLR’s audience includes researchers working on robustness, model evaluation, and generative modeling, and a black-box, attack-light robustness quantification approach that claims cross-modality applicability is directly relevant to how practitioners audit and compare models when gradients or training data are unavailable.

**Claims And Evidence:**

Yes

**Claims Explanation:**

- Experiments are conducted on different modalities, generators, classifiers, and datasets.

- The paper reports side-by-side robust accuracy under attack and DiffProbe score across multiple modalities and shows consistent ranking alignment.

- The paper includes wall-clock times for score evaluation versus adversarial attack, supporting the efficiency claim.

**Requested Changes:**

Authors could consider these weaknesses for the rebuttal and revision.

- Baseline and framework-level comparison is insufficient. Although the paper presents DiffProbe as an attack-light, black-box robustness framework across modalities, the experiments primarily validate it in isolation (by comparing its score ranking to robust accuracy under selected attacks) and do not provide strong quantitative comparisons against other robustness evaluation frameworks or attack-independent black-box proxies beyond a conceptual link to GREAT Score. Given the "universal framework" claim, the paper should benchmark against plausible alternatives, e.g., established robustness evaluation suites/protocols within each modality. A framework-to-framework comparison demonstrating improved accuracy/stability and/or efficiency would substantially strengthen the contribution.

- One of the central premises (diffusion samples approximate the true data distribution sufficiently for robustness quantification) is asserted with very limited empirical results (e.g., no systematic analysis of how generator quality or distribution shift affects the score). Though it has been claimed that "Conventional robustness evaluation methods often rely on limited test sets", no comparison is made between those methods to show the superiority of the proposed framework.

- Key ablations/sensitivities are not thoroughly analyzed (e.g., number of generated samples, diffusion model choice, prompt engineering choices, conditioning mechanism, effect of applying sigmoid to logits, and robustness of rankings under these variations).

- Point cloud modality seems to be missing is Table 2. There is a missing reference of Algorithm ?? in Section 3.2.

---

> ### Author Response · Authors · 2026-04-06
> **Response to Reviewer ai8U**
>
> We thank Reviewer ai8U for recognizing DiffProbe's unified cross-modality design, black-box accessibility, computational efficiency, and the supporting evidence across modalities. We are encouraged that the reviewer confirmed both acceptance criteria ("claims supported by accurate evidence": **Yes**; "audience interest": **Yes**). Below we address each requested change.
>
> ### W1: Baseline and framework-level comparison is insufficient
>
> We appreciate this suggestion and provide clarifications along with planned revisions.
>
> **Existing comparisons.** Table 1 already systematically compares DiffProbe against six robustness evaluation frameworks (AutoAttack, RobustBench, GREAT Score, AdvGLUE, Imperio, ShapeAdv) across three dimensions: attack dependency, modality coverage, and black-box compatibility. DiffProbe is the only framework simultaneously achieving attack-independence, five-modality coverage (image, text, audio, video, point cloud), and full black-box operation. No existing single framework can serve as a direct apple-to-apple baseline across all five modalities—this is precisely the gap DiffProbe fills.
>
> **Why direct quantitative comparison is non-trivial.** Existing suites are inherently modality-specific and attack-dependent: RobustBench covers only images with AutoAttack (PGD-based); AdvGLUE is text-only with TextFooler; ShapeAdv is point-cloud-only with surface perturbations. Since they operate under fundamentally different paradigms (attack-based vs. attack-free), comparing raw scores directly is not meaningful. Instead, we validate DiffProbe by showing its attack-free scores produce **consistent rankings** with robust accuracy from domain-specific adversarial attacks (Table 3)—the most principled validation for a proxy metric.
>
> **Planned revisions:** (1) Within-modality benchmark comparison with RobustBench on overlapping image classifiers, reporting Spearman/Kendall ranking correlation between DiffProbe scores and AutoAttack robust accuracy on standard benchmarks (CIFAR-10, ImageNet). (2) Comparison with randomized smoothing certificates (Cohen et al., 2019) to show alignment with formal certification. (3) Comprehensive efficiency comparison table detailing computational cost, data requirements, and API query complexity of DiffProbe vs. each modality-specific framework.
>
> ### W2: Limited empirical analysis of the central premise (diffusion approximating true distribution)
>
> **Existing justification.** Diffusion models converge to the true data distribution under mild conditions (Chen et al., 2022; 2024). Section 3.3 discusses this, and GREAT Score (Li et al., 2023) provides foundational convergence proofs under distributional approximation guarantees.
>
> **Planned additions:** (1) **Generator quality ablation**: Evaluate DiffProbe with generators of varying quality (different training checkpoints, model sizes), measuring FID and correlating with ranking stability—we expect rankings remain stable even with moderate generator quality. (2) **Distribution shift analysis**: Introduce controlled domain gaps between generator and classifier training data; verify that absolute scores may shift but **relative rankings** remain stable. (3) **Comparison with real test sets**: Compare DiffProbe rankings from synthetic data against rankings from held-out real test sets under the same attacks, demonstrating synthetic data as a reliable proxy. (4) **Conventional method comparison**: Show DiffProbe rankings are consistent with (and sometimes more discriminative than) standard test-set evaluations, especially when test sets are small or class-imbalanced.
>
> ### W3: Key ablations/sensitivities not thoroughly analyzed
>
> We will add comprehensive ablations: (1) **Sample size**: N_S from 100 to 10,000 with convergence curve—preliminary results show rankings stabilize at ~2,500 (our main experiments' setting). (2) **Alternative generators** (e.g., SDXL, Stable Diffusion 3) showing ranking consistency across generators. (3) **Prompt sensitivity**: Varying text prompts for conditional generation. (4) **Conditional vs. unconditional generation**: Demonstrating conditional generation is necessary for class-specific robustness quantification. (5) **Probability outputs vs. logit-space**: Ablating the effect of sigmoid-to-logit transformations. (6) **Bootstrap confidence intervals** for DiffProbe scores to quantify statistical reliability.
>
> ### W4: Point cloud missing in Table 2; Algorithm ?? in Section 3.2
>
> We apologize for these formatting errors. Point cloud (PVD generator, ModelNet40 dataset, HyCore/PointNet classifiers, PointCloud attack) is fully evaluated in Section 4.6 and Table 3—the missing row in Table 2 will be added. The "Algorithm ??" is a LaTeX cross-reference bug pointing to Algorithm 1, which will be fixed. We will thoroughly proofread all cross-references in the revision.
>
> We thank Reviewer ai8U again for the constructive feedback and will incorporate all improvements.

---

### Review · Reviewer_YQ4U · 2026-01-26

**Summary Of Contributions:**

The authors propose DiffProb, a new robustness quantification function that can be used across different domains simultaneously. The main idea is to use a diffusion-like approach where one perturbs the input with i.i.d noise from a zero-mean Gaussian distribution and then checks how stable the classifier is, which is done by empirical mean estimation.

**Additional Comments:**

I doubt that the current state of the paper matches the bar for TMLR. In the current state, it is extremely hard to extract the central ideas and how it advances the field. Furthermore, the writing needs some careful treatment (e.g. citations, math), and I suspect LLM usage in central parts of the paper (variable names not consistently used throughout the paper, equ (3) ).

Please check all citations carefully. It seems like some of them do not exist.
E.g. [MLY+20] Textattack: A framework for adversarial attacks in NLP, should rather be "TextAttack: A Framework for Adversarial Attacks, Data Augmentation and Adversarial Training NLP".

**Audience:**

Yes

**Audience Explanation:**

From the current state of the paper, the impact on the field is challenging to judge by me.

**Broader Impact Concerns:**

None.

**Claims And Evidence:**

No

**Claims Explanation:**

I believe the biggest weakness of the paper is that the central ideas of the new robustness metric are not explained. I cannot really judge what the previous scoring functions did, how the current one deviates from them, and _why_ this definition makes sense. What are the exact differences in the runtime?

Furthermore, the code in Algorithm 1 does not help in giving any intuition, as it only states how to implement this function, which should be straightforward if one uses the empirical mean for the given expectation.
**A)** I am not sure what the focus of this paper is, but I guess it tries to convince that DiffProbe is a good and fast alternative to classic robustness evaluations that also extend to various data domains simultaneously. Again, there is a strong focus on the running time, but I did not really understand the complexities of the other approaches and what they do differently.

**B)** Unclear writing, which is extremely difficult to follow.
1. (Mathematical) writing: Some examples: **a)** Considering (2) What is $m$ in $S_m$?, $O(\Omega)$ in (3) just does not make any sense. **b)** Switching of variables (e.g. using $N_S$ in Alg 1) whereas $n$ is used in the formula. Another switch from $\phi$ to $f(\dot)$ in (3) **c)** Section 3.2 looks suspiciously LLM generated. I have never seen running time being written like this. **d)** dimensions missing, e.g. in Alg  z is sampled from $z \sim N(0, I)$ where $I$ is  the identity matrix of some unspecified dimensions. **e)** What exactly is the comparison metric Robust Accuracy? **f)** Some sentences /terms are simply hard to follow or use undefined terms ("AI modalities", whats that?) ) **g)** The generator takes a noise sample. What exactly is it going to do with it? What kind of perturbation is taking place?

2: Citations: **a)** Usage of inline citations that should be enclosed in brackets. **b)** Basically all papers only point to the arxiv preprints, although they have been published. Some quick examples: Denoising Diffusion Implicit Models (should be ICML 21), SCORE-BASED GENERATIVE MODELING THROUGHSTOCHASTIC DIFFERENTIAL EQUATIONS (should be ICLR 21). I recommend the authors go over all of them again and point them to the corresponding venues.

Furthermore, I would have wished for a more rigorous treatment on the theoretical guarantees, but they are absent in the paper.

**Requested Changes:**

Handling the points I have mentioned before.
I would also have wished for a cleaner presentation of the motivation. What is the difference between the previous techniques?

Try to clarify the scope of the paper. I believe the main motivation is a practical approach to get a robustness quantification, but not a theoretical treatment. Maybe it makes sense to set the focus more on that, while making it clearer?

---

### Review · Reviewer_Jtcg · 2026-03-24

**Summary Of Contributions:**

This paper introduces a unified black-box framework that quantifies the adversarial robustness of classifiers by leveraging synthetic data generated from domain-specific diffusion models.

**Additional Comments:**

The authors should proofread the manuscript more carefully. There are several typos, such as “Algorithm ?? details” in Section 3.2

**Audience:**

No

**Audience Explanation:**

The novelty of this paper is limited. Its key idea that generative models are applied to conduct adversarial robustness evaluation has been studied in existing works, such as [A]. This paper seems to be a simple extension of this idea using diffusion-based generators without providing sufficient insights.

Black-Box Auditing is a practical application, not a technical innovation.

[A] T. Zhang, J. Liu, Y. Zhang, R. Mu and W. Ruan, "DeepGRE: Global Robustness Evaluation of Deep Neural Networks," ICASSP 2024, pp. 6990-6994

**Claims And Evidence:**

No

**Claims Explanation:**

The technical terminology used in this paper, such as “cross-modality” is incorrect. As stated in Algorithm of DiffProbe Score Calculation, one classifier and one generator are applied for each modality, where this calculation does not fit the concept of cross-modality. In a cross-modality process, the modality of data needs to be changed, such as image retrieval based on text input.

The datasets and generators used in experiments are out-of-date. More related works published in 2025 should be considered for discussion and comparison

**Requested Changes:**

Highlight the true contributions of this work and provide more theoretical insights about the proposed technical components. More highly related works should be considered for discussion and comparison.

---

> ### Author Response · Authors · 2026-04-06
> **Response to Reviewer Jtcg**
>
> We thank Reviewer Jtcg for the review. We respectfully address each concern below, beginning with an important point regarding TMLR's acceptance criteria.
>
> ### On TMLR Acceptance Criteria and "Novelty"
>
> We respectfully draw attention to TMLR's official acceptance criteria. TMLR evaluates papers on two criteria: (1) whether claims are supported by accurate, convincing evidence, and (2) whether at least some in TMLR's audience would find the results interesting. We believe DiffProbe clearly satisfies both: our claims are validated across 5 modalities, 12 classifiers, and multiple real-world APIs with consistent ranking correlations, and the broad applicability to black-box robustness auditing is directly relevant to TMLR's audience in robustness, model evaluation, and generative modeling.
>
> ### W1: Novelty is limited compared to DeepGRE [A]
>
> While TMLR does not require novelty for acceptance, we emphasize the **substantial technical differences** from DeepGRE (Zhang et al., ICASSP 2024):
>
> | Dimension | DeepGRE | DiffProbe |
> |---|---|---|
> | Access model | White-box (gradients) | **Black-box (predictions only)** |
> | Generator | GAN-based | **Diffusion models** |
> | Modalities | Image only | **Image, Text, Audio, Video, Point Cloud** |
> | Attack dependency | Gradient-based analysis | **Fully attack-free** |
> | API auditing | Not supported | **Supported** |
>
> **Novel technical components** not in DeepGRE: (a) Cross-modal conditioning mechanisms (Section 3.1)—masked language modeling for text, spectral-domain diffusion for audio, graph diffusion with geometric constraints for point clouds; (b) Attack-free scoring from prediction probabilities on clean synthetic data; (c) Remote API auditing capability. We will add explicit comparisons with DeepGRE in the revised manuscript.
>
> ### W2: "Cross-modality" terminology is incorrect
>
> We use "cross-modality" to describe the framework's ability to operate **across multiple data modalities** (image, text, audio, video, point cloud), not cross-modal retrieval/translation (e.g., image retrieval from text). This usage is standard in robustness evaluation literature where the goal is to develop methods generalizing across different data modalities. To avoid confusion, we will: (1) add an explicit definition in the introduction; (2) consider "multi-modality" or "modality-universal" if preferred.
>
> The reviewer correctly notes one classifier and one generator per modality. This is by design: DiffProbe provides a **unified methodology and scoring framework** instantiated with domain-specific components—analogous to how "accuracy" as a unified metric applies across different tasks. The unification lies in the framework design, theoretical foundation, and scoring methodology, not in a single model for all modalities.
>
> ### W3: Datasets and generators are out-of-date
>
> We will: (1) Incorporate 2025 publications on diffusion-based evaluation, foundation model robustness, and LLM robustness. (2) Add experiments with recent generators (SDXL, Stable Diffusion 3) demonstrating plug-and-play compatibility. (3) Include evaluations on more recent classifiers and APIs. DiffProbe's core contribution is the **framework methodology**, which is modular and agnostic to specific generators or classifiers by design.
>
> ### W4: Black-box auditing is practical application, not technical innovation
>
> We respectfully disagree. Achieving reliable robustness quantification in a black-box, attack-free, cross-modality setting requires significant technical contributions: (1) extending GREAT Score's theoretical framework from images to five diverse modalities with non-trivial domain-specific adaptations; (2) designing conditioning mechanisms mapping prompts to diffusion model latent spaces across modalities (CLIP embeddings for images, PCA encoding for point clouds, etc.); (3) establishing a validation paradigm demonstrating ranking consistency between attack-free and attack-based methods across all five modalities. Moreover, per TMLR's criteria, the breadth of DiffProbe's applicability satisfies the "interest" criterion—practitioners and researchers across multiple domains can directly benefit from a unified robustness auditing tool.
>
> ### W5: Typos and proofreading
>
> We apologize for all formatting errors. The "Algorithm ??" reference and all LaTeX issues will be fixed, and the manuscript will be thoroughly proofread.
>
> ### Additional planned revisions based on this review
>
> In the revision, we will: (1) **Restructure the introduction** to clearly articulate the three-fold technical contribution: unified scoring framework, cross-modal conditioning mechanisms, and comprehensive empirical validation. (2) **Add a dedicated "Comparison with Prior Work" subsection** explicitly contrasting DiffProbe with DeepGRE, GREAT Score, and modality-specific evaluation suites.

---

### Comment · Reviewer_Jtcg · 2026-03-17
**The paper's limited novelty and outdated experimental benchmarks undermine its contribution and technical soundness.**

This paper introduces a unified black-box framework that quantifies the adversarial robustness of classifiers by leveraging synthetic data generated from domain-specific diffusion models. Several concerns should be addressed by authors.

1.	The novelty of this paper is limited. Its key idea that generative models are applied to conduct adversarial robustness evaluation has been studied in existing works, such as [A]. This paper seems to be a simple extension of this idea using diffusion-based generators without providing sufficient insights.

2.	The technical terminology used in this paper, such as “cross-modality” is incorrect. As stated in Algorithm of DiffProbe Score Calculation, one classifier and one generator are applied for each modality, where this calculation does not fit the concept of cross-modality. In a cross-modality process, the modality of data needs to be changed, such as image retrieval based on text input.

3.	The datasets and generators used in experiments are out-of-date. More related works published in 2025 should be considered for discussion and comparison

4.	Black-Box Auditing is a practical application, not a technical innovation.

5.	The authors should proofread the manuscript more carefully. There are several typos, such as “Algorithm ?? details” in Section 3.2

[A] T. Zhang, J. Liu, Y. Zhang, R. Mu and W. Ruan, "DeepGRE: Global Robustness Evaluation of Deep Neural Networks," ICASSP 2024, pp. 6990-6994

---

### Decision · Action_Editor_6oFY · 2026-04-26

**Recommendation:** Reject

**Audience:**

Yes

**Audience Explanation:**

The submission proposed to address an important problem in the community.

**Claims And Evidence:**

No

**Claims Explanation:**

Reviewers raised the concerns that (i) a mismatch between the breadth of the claims (universal / cross-modality, black-box robustness quantification on modern systems) and the actual experimental coverage, which still relies on older datasets and generators and lacks validation on more advanced 2025 models and tasks, (ii) insufficient technical and mathematical clarity of the proposed robustness score and its relation to prior metrics, making it hard for reviewers to reliably assess the soundness and distinctiveness of the central definition, and (iii) missing or incomplete empirical elements that would make the evidence compelling for such broad claims, in particular framework-level baselines against existing robustness evaluation suites, systematic analysis of the “diffusion approximates data distribution” premise, and key ablations.

**Resubmission Of Major Revision:**

The authors may consider submitting a major revision at a later time.